# Novel Meta-Learning Techniques for the Multiclass Image Classification Problem

**DOI:** 10.3390/s23010009

**Published:** 2022-12-20

**Authors:** Antonios Vogiatzis, Stavros Orfanoudakis, Georgios Chalkiadakis, Konstantia Moirogiorgou, Michalis Zervakis

**Affiliations:** School of ECE, Technical University of Crete, 731 00 Chania, Greece

**Keywords:** ensemble learning, mixture of experts, decomposition-based methods, multi-class classification, Bayes rule, opinion aggregation, meta-learning

## Abstract

Multiclass image classification is a complex task that has been thoroughly investigated in the past. Decomposition-based strategies are commonly employed to address it. Typically, these methods divide the original problem into smaller, potentially simpler problems, allowing the application of numerous well-established learning algorithms that may not apply directly to the original task. This work focuses on the efficiency of decomposition-based methods and proposes several improvements to the meta-learning level. In this paper, four methods for optimizing the ensemble phase of multiclass classification are introduced. The first demonstrates that employing a mixture of experts scheme can drastically reduce the number of operations in the training phase by eliminating redundant learning processes in decomposition-based techniques for multiclass problems. The second technique for combining learner-based outcomes relies on Bayes’ theorem. Combining the Bayes rule with arbitrary decompositions reduces training complexity relative to the number of classifiers even further. Two additional methods are also proposed for increasing the final classification accuracy by decomposing the initial task into smaller ones and ensembling the output of the base learners along with that of a multiclass classifier. Finally, the proposed novel meta-learning techniques are evaluated on four distinct datasets of varying classification difficulty. In every case, the proposed methods present a substantial accuracy improvement over existing traditional image classification techniques.

## 1. Introduction

Multiclass classification aims to assign instances of data to one of three or more classes. In a conventional learning process for multiclass classification, considering that there are k>2 classes, i.e., $Y=[C_{1},C_{2},\ldots ,C_{k}]$, and *n* training instances, i.e., $S=\{\left(x_{1},y_{1}\right),\left(x_{2},y_{2}\right),\ldots ,\left(x_{n},y_{n}\right)\}$, each training instance belongs to one of *k* distinct classes, and the objective is to build a function f(x) that, given a new data instance *x*, can accurately predict the class to which the new instance belongs.

In the real world, image classification [1], text classification [2], e-commerce product categorization [3], medical diagnosis [4], and other multiclass classification challenges are quite prevalent. Multiclass decomposition separates a multiclass classification problem into a group of independent binary learners and re-composes them by combining their outputs to reconstruct the multiclass classification results.

There are numerous concrete representations of decomposition methods, such as one-vs.-rest (OvR) [5] and one-vs.-one (OvO) [6]. Notably, OvR trains *k* unique base learners for the *i*th of which the positive examples are all cases in class $C_{i}$ and the negative examples are all instances not in $C_{i}$; OvO trains k(k−1)/2 base learners, one for each pair of classes. Even while OvR and OvO are straightforward to implement and widely used during practice, they yield several apparent drawbacks.

First, both OvR and OvO are based on the assumption that all classes and their corresponding base learners are independent, ignoring the latent association between these classes in practical situations. For example, in an image classification task for waste recycling, instances under the “paper” class appear to have a higher association with instances under the “cardboard” class than instances under the “plastic” class. In addition, the training of OvR and OvO is ineffective because of their high computational complexity when *k* is significantly large, which results in a prohibitively expensive training cost when processing large-scale classification datasets [7]. To alleviate and exploit some of the shortcomings mentioned above, in this work, meta-learning approaches are employed to combine the outputs of the base learner more efficiently to obtain a superior outcome.

Several implementations solve this multiclass classification problem, each taking a different approach in terms of algorithms or how they integrate data from various networks under the umbrella of the ensemble learning family of techniques. Initially, some methods follow a decomposition strategy, employing smart criteria to separate the information [8,9] or various mathematical tricks to find the most suitable binary classifier [7,10]. There is also the issue of information transferability with each new piece of data or when combining networks. We should not forget that there are multiple approaches applied at the level of decision rules [11,12,13], known as ensemble techniques, which aggregate opinions from different methods or networks into clusters with parallel training [14]. Finally, other studies focus on combining expert techniques that utilize a generalizer to coordinate information allocation for the ultimate decision [15,16].

In this research, we examine the topic of image classification using deep learning to determine appropriate image classification algorithms for multiclass problems. Specifically, we put forward several classification techniques based on ensembles of deep neural networks, trying to increase the accuracy of the multiclass classification problem. We examine and assess several meta-learning techniques/approaches. Each multiclass problem can be divided into smaller binary problems to have a better separation between classes and distinct boundaries with regard to the features of each class.

In general, we consider an ensemble composed of the primary multiclass classifier supplemented by individual components trained with a focus on a single class (OvR). The application novelty of this approach is that we can blend various multicenter datasets with a diverse focus on the material of interest in a single ensemble scheme.

Classifiers are trained in two stages, one for individual classifiers and the second for decision aggregation. The second layer may be implemented using gating functions (with a Bayesian formulation) or as a meta-learning (neural network) technique trained on a dataset containing the primary classes.

Our main contributions are the following:A series of novel approaches for combining the output of a decomposed multiclass (image) classification task under the umbrella of the one-vs.-rest framework for the opinion aggregation module.A novel opinion aggregation method that combines information derived from sub-class classifiers based on the Bayes theorem. The novelty lies within two key parts:
−The first part is the usage of multiple unrelated multicenter datasets to train the expert sub-class classifiers.−Secondly, on the way, it calculates the inputs of the final meta-learner level using the baseline multiclass classifier along with the output of the expert sub-class classifiers using the Bayes theorem.A novel design for the mixture of expert approaches that incorporates the knowledge of a multiclass classifier as a gating model.In addition, we put forward two stack generalization variants with novel characteristics that follow the one-vs.-rest architectural paradigm:
−The first one divides the initial problem into *n* separate classifiers and uses a generalizer to learn the optimal weight polic y.−The second one is similar to the first, but it additionally feeds the generalizer with the output of the baseline multiclass classifier.We expand on the concept of shared wisdom from data and explore how various datasets (such as hierarchical labeled datasets or even simply related datasets) can be combined to improve accuracy and create a stable network architecture.The methods proposed in this study can also be used to train transfer learning models for even more accurate and generalized classifiers or even for different learning tasks.Last but not least, this framework can be used to apply a series of black-box optimizations, potentially with varying weights and parameters and based on various intuitions and scenarios of interest.

The remaining sections of this work are structured as follows. In Section 2, we review related research and provide the necessary background. In Section 3, the meta-learning phase of the proposed methodology is described, which employs a one-vs.-rest decomposition method with diverse opinion aggregation mechanisms. The experimental design of this research is then described in Section 4.1. The experimental results are reported in Section 4.2, while conclusions and future work are discussed in Section 5.

## 2. Background

In this section, the necessary background information is provided. In detail, several multiclass classification learning techniques are presented. We are starting by introducing traditional ensemble learning techniques, followed by voting ensembles and more sophisticated extensions of voting ensemble learning, mixture of experts techniques, and stacked generalization variants. Finally, in the last section, we outline significant developments and related work in multiclass classification utilizing machine learning algorithms, most notably CNN techniques, to offer context for the current work in this area.

### 2.1. Ensemble Learning

Ensemble learning is a technique for constructing and combining multiple models, such as classifiers or experts, to solve a particular computational intelligence problem [17]. Typically, ensemble learning is employed to improve the performance of a model (classification, prediction, feature approximation, etc.) or to counteract the tendency of a weak model. Ensemble learning can also be used to assign confidence to a model’s decision features, combine data, conduct incremental learning, non-stationary learning, and error correction, and select optimal (or near-optimal) features [18]. While this article focuses on ensemble learning applications for classification, many of the fundamental ideas discussed can be easily applied to approximation and prediction-related topics.

An ensemble-based design is achieved by combining many models, henceforth “classifiers”. These systems are, therefore, sometimes referred to as multiple classifier systems or ensemble systems [19]. Several situations in which ensemble-based methods make mathematical sense are elaborated on in this section. A real-world analogy of ensemble learning is the procedure before a critical medical assessment. Patients typically seek the advice of multiple physicians before committing to a crucial medical procedure, read customer reviews before purchasing a product (especially an expensive plane ticket), and evaluate potential employees by reading their references, among other practices. In each situation, the final evaluation will be determined by weighing the diverse perspectives of a group of experts. The primary objective is to prevent the undesirable possibility of an unnecessary medical procedure, a defective product, or an unskilled employee.

### 2.2. Voting Ensembles

The voting ensemble (or “majority voting ensemble”) is a machine learning ensemble that combines several models’ predictions. It is a technique that can be employed to improve model performance, particularly to the point where it outperforms all other models in the ensemble. The ensemble voting algorithm combines the predictions of multiple models. It is appropriate for classification and regression tasks. In regression, this involves estimating the average sample predictions. In terms of classification, the guesses for each category are added together, and the label with the most votes is selected.

The voting ensemble technique is a potent instrument that comes in handy when a single model is biased [20,21,22]. In addition, the voting ensemble may produce a higher overall score than the best base estimator because it combines the predictions of multiple models and attempts to compensate for the flaws of the individual models. Diversifying the base estimators as much as possible is one method for increasing the efficiency of the ensemble. As depicted in Figure 1, the base learners will be distinct pre-trained, fine-tuned models using the same dataset.

Voting classifiers typically employ two distinct voting methods:**Hard Voting:** Every classifier will vote for a particular class, and the majority will prevail. In mathematical terms, the desired target label of a set corresponds to the mode of the distribution of independently predicted labels.**Soft Voting:** Each classifier assigns to each data point a probability that it belongs to a particular target class. Predictions are summarized and weighted based on the value of the classifier. Then, the vote is cast for the preferred label with the highest probability after weighting.

Ensemble voting does not ensure that its results will be superior to any other model utilized by the ensemble. If any proposed method outperforms the voting ensemble, it is assumed that the outperforming method will be adopted. A voting ensemble is incredibly beneficial for machine learning models that use stochastic learning methods and generate a unique final model each time they are trained on the same dataset. For example, neural networks utilize stochastic gradient descent to identify the optimal solution. When multiple instances of the same machine learning algorithm are combined with slightly different hyperparameters, voting ensembles are also particularly effective.

### 2.3. Meta-Learning Techniques: Stacked Generalization

A drawback of the voting framework is that each model must contribute equally to the final prediction. This could be a problem if some models perform poorly in some conditions but admirably in others. To address this issue, the literature suggests a voting ensemble extension employing a weighted average or a weighted voting system for the contributing models. This is typically referred to as “*blending*” [23]. When using a machine learning model to determine how much to assist each model when making predictions is yet another expansion. This is referred as “*stacked generalization*”, or “*stacking*” [24] for short.

A further generalization of this method involves substituting any learning technique for the linear weighted sum (e.g., a linear regression) model used to integrate the predictions of the sub-models, specifically by training a brand-new model to determine the optimal method for combining the contributions of each sub-model [25].

In contrast to bagging or boosting, which trains multiple versions of the same learner, stacking (or stacked generalization) creates a series of models using a variety of learning methodologies (e.g., one neural network, one decision tree, all decision trees). For example, think of the scenario where *m* models are trained on a dataset of *n* samples. We intend to train *m* binary classifiers $h_{j}$ sequentially to combine them later to select new instances *x* as their weighted majority vote.

The model outputs are used to calculate the final prediction for any instance x:(1)$$\hat{y}\left(x\right)=\sum_{j=1}^{m}w_{j}\cdot h_{j}\left(x\right)$$
where the Level-1 meta-learner (e.g., a neural network) optimizes the weights $w_{i}$ of the Level-0 base learners. That is, the individual *m* predictions associated with each training instance $x_{i}$ are forwarded as training data to the Level-1 learner, as shown in Figure 2.

Meta-learning, in general, refers to algorithms that acquire knowledge from other learning algorithms. This often entails employing machine learning algorithms that learn how to optimally aggregate the predictions of other machine learning algorithms in the field of ensemble learning. However, meta-learning may also refer to a researcher’s human model selection and algorithm tuning on a machine learning task, which current auto-ml [26] algorithms strive to automate.

A simple method for combining multiple models under an ensemble scheme is calculating the mean of all sub-model outputs. *Model averaging* is an ensemble technique in which multiple sub-models contribute equally to an aggregated prediction. Model averaging can be enhanced by weighting the contributions of each sub-model to the aggregate prediction according to the individual performance of the sub-models. A non-weighted model averaging ensemble aggregates the predictions of various trained models. This strategy has the shortcoming that each model contributes the same proportion to the ensemble prediction, regardless of its performance. This method’s variant, the weighted average ensemble, weights each ensemble member’s contribution based on the model’s confidence or expected performance on a holdout dataset. This enables models with better performance to contribute more, while models with poorer performance contribute less. The weighted average ensemble typically outperforms the model’s average ensemble when the weights can be configured accurately.

### 2.4. Mixture of Experts

The mixture of experts (MoE) is a traditional ensemble architecture in which each expert is specialized in a specific subset of the input space or expertise domain. In this way, it is hoped to specialize experts on minor challenges, thereby resolving the original issue via a divide-and-conquer approach. MOE architecture is composed of *N* networks of experts. These experts are combined via a gating network that divides the input space proportionally. It employs a divide-and-conquer strategy managed by a gating network. Using a specialized cost function, the experts specialize in their respective subspaces. Utilizing experts’ discriminative ability is superior to clustering. The gating network must determine how to distribute examples to different specialists. Such models have the potential to ease the development of more extensive networks that are inexpensive to compute during testing and more parallelizable during training. Precisely, the strategy consists of four key elements:**A task is divided into sub-tasks:** The first step is to break the predictive modeling problem into smaller chunks. This frequently entails applying domain knowledge to the key problem, determining the task’s natural division, and then deriving an effective approach from sub-solutions.**Develop an expert for each sub-task:** Experts are typically neural network models that predict a numerical value in regression or a class label in classification because the mixture of experts approach was initially developed and studied in the field of artificial neural networks. Each expert is provided with an identical input pattern and tasked with making a prediction.**Utilize a gating model to decide which expert to consult:** The gating network receives the input pattern that was provided to the expert models and outputs the allocation that each expert should make when generating a prediction for the input. Because the MoE effectively learns which portion of the feature space is learned by each ensemble member, the weights generated by the gating network are assigned dynamically based on the input. Simultaneously training the gating network and the experts in neural network models enables the gating network to determine when to trust each expert’s prediction. Traditionally, this training method employed expectation maximization (EM). A “softmax” output from the gating network might provide each expert with a confidence score that is similar to a probability score.**Pool predictions and gating model output to make a prediction:** Lastly, a prediction should be made using a pooling or aggregation strategy, which is carried out by combining expert models. Selecting the expert with the highest output or confidence provided by the gating network could be as straightforward as that. Alternately, a weighted sum prediction that explicitly incorporates each expert’s prediction and the gating network’s confidence estimation could be constructed.

The training strategy generally seeks to accomplish two objectives: identifying the optimal gating function for a given set of experts and instructing the experts on the distribution indicated by the gating function.

### 2.5. Related Work

In this subsection, we explain several advancements in multiclass classification using machine learning algorithms, most notably CNN approaches, to provide context for the current related work. The works that will be presented are divided into four categories: (a) multiclass decomposing scheme, (b) incremental learning, (c) optimization techniques exploiting decision ruling on the outcome of primary classifiers, and (d) mixture of experts.

#### 2.5.1. Multiclass Decomposition Techniques

Decision tree algorithms have been shown to be an effective technique in classification challenges. However, their classification performance is inadequate in multiclass contexts [27]. In a study [8], decision tree algorithms are integrated with the one-vs.-rest (OvR) binarization technique to increase the scheme’s generalization capabilities. In contrast to prior research, which focused on aggregation methodologies, [8] focused on the process of developing base classifiers for the OvR scheme. The proposed method combines distribution information with permutation information derived from training data for each split point at the root or internal nodes. The initial discussion focuses on the Hellinger distance. The permutation information is then analyzed, and a new concept of split ratio is presented. The optimal splitting point is determined using the following principle: if a node contains more information about splitting points, it may be easier to make the right decision. In this context, a unique split criterion referred to as the splitting point correction matrix (SPCM) is presented, which can successfully address the unbalance issue created by the OvR scheme.

The central argument of the survey “In Defense of One-vs.-Rest Classification” [7] is that a simple “one-vs.-rest” methodology is as accurate as any other method, provided that the underlying binary classifiers are well-tuned regularized classifiers, such as support vector machines. Using the best binary classifier available is, according to [7], the most crucial aspect of multiclass classification. Once this has been accomplished, it appears to make little difference which multiclass scheme is implemented; hence, a simple system such as OvR is preferred over a more complex error-correcting coding scheme or single-machine approach.

In [9], the authors develop a one-vs.-one (OvO) training process for neural networks that teaches each output unit to differentiate between a specific pair of classes. Moreover, compared to the one-vs.-rest categorization system, this method produces more output units. The proposed architecture has an output layer with K∗(K−1)/2 output units and a shared feature learning component. Each output is trained to distinguish between inputs of two classes and ignore examples of other classes. They devised three steps to develop the OvO classification scheme: (a) creating a code matrix to transform the one-hot encoding to a new label encoding, (b) altering the output layer and loss function, and (c) altering the classification method for new (test) examples. To determine the benefits of the proposed method, they compared it to the outcomes of one-vs.-one and one-vs.-rest classifiers on three distinct plant recognition datasets and a dataset including photographs of multiple monkey classes. Two deep convolutional neural networks (CNN) architectures were constructed from scratch or using pre-trained weights: Inception-V3 and ResNet-50. It is reported in [9] that the one-vs.-one classification outperforms the one-vs.-rest classification when all CNNs are constructed from scratch. However, fine-tuning the two pre-trained CNNs using the one-vs.-rest method yields the best results, as each CNN was previously fine-tuned using this method.

#### 2.5.2. Continual Learning/Incremental Learning

Vogiatzis et al. [11] introduce a novel image classification model that can efficiently differentiate among recyclable materials. They present the so-called “Dual-branch Multi-output CNN”, a customizable convolutional neural network with two branches designed to (i) classify recyclables and (ii) further detect the type of plastic. The proposed architecture consists of two classifiers trained on two distinct datasets to encode complementary properties of recyclable materials. Densenet121, ResNet50, and VGG16 networks were used in the Trashnet dataset with data augmentation techniques and the WaDaBa dataset with physical variation approaches in their research. Specifically, their method uses the joint utilization of the datasets, enabling the learning of disjoint label combinations. Experiments have demonstrated its efficacy in waste classification. Additionally, another research study expands on the techniques developed in the work mentioned above and proposes an autonomous, intelligent robotic system for categorizing and separating recyclable materials, aiming to contribute to increasing the recycling rates in Greece. Specifically, ref. [28] introduces a two-classifier incremental learning scheme with the first model trained on RGB waste images and the second trained on near-infrared spectrum waste images. Another similar study [12] found that the D-LinkNet architecture, first introduced for road segmentation, can outperform other implementations for classifying power line structures. In detail, D-LinkNet adopts an encoder–decoder structure, a dilated convolution on a pre-trained encoder.

With memory-resource-limited limitations, class-incremental learning (CIL) typically encounters the “catastrophic forgetting” issue when updating the joint classification model in response to adding new classes. To address the forgetting problem, numerous CIL approaches transfer the knowledge of old classes by storing certain representative samples in a memory buffer with limited capacity [13]. To better use the memory buffer, ref. [13] proposes storing more auxiliary low-fidelity exemplar samples, as opposed to the original ones. This memory-efficient exemplar preservation approach makes the transmission of old-class information more efficient. However, the low-fidelity exemplar samples are frequently dispersed in a different domain than the original exemplar samples; this phenomenon is known as a domain shift. To overcome this limitation, ref. [13] presents a duplet learning approach that aims to build domain-compatible feature extractors and classifiers, therefore significantly reducing the aforementioned domain gap. Consequently, these low-fidelity auxiliary exemplar samples can partially replace the original exemplar samples at a reduced memory cost. In addition, they provide a robust classifier adaptation strategy that refines the biased classifier (trained using samples including distillation label knowledge about old classes) using samples with pure, actual class labels.

#### 2.5.3. Opinion Aggregation Techniques on the Final Level

In Hinata and Takahashi’s work [29], a complex network technique referred to as EnsNet is developed for classification purposes. EnsNet comprises a primary CNN and several fully connected sub-networks. In the final convolutional layer, all sub-networks of the basic CNN construct a feature map, and a majority vote determines the final class selection. In the final layer, all sub-networks participate in an ensemble despite training independently. The set of feature maps created by the last convolutional layer of the base CNN is partitioned along channels into disjoint subsets, with each subset being passed into one of the sub-networks. Each sub-network consists of a fully interconnected neural network with several weight layers. Experiments have demonstrated that that is the model with the lowest error rate compared to other cutting-edge models.

In another study [14], the authors propose a generic classification architecture of independent parallel CNNs that explicitly exploits a “mutual exclusivity” or the “classifiers’ mutually supported decisions” property underlying many dataset domains of interest, namely that in many instances, an image in a given dataset may belong to only one class. The proposed system consists of numerous opinion aggregation decision rules that are activated when the mutual exclusivity property is or is not satisfied, as well as weights that intuitively reflect the confidence that each CNN identifies its related class. Thus, this approach can: (a) take advantage of obviously identified class characteristics when they exist and (b) confidently assign objects to classes even when class boundaries are unclear.

In the past, there have been excessive studies to determine which method is better comparing different classification algorithms in the binarization strategies domain. In this paper [10], researchers are interested in ensemble methods based on binarization techniques; more specifically, the authors focus on the well-known one-vs.-one and one-vs.-rest decomposition strategies, paying particular attention to the final step of ensembles and the combination of the outputs of the binary classifiers. To combine these outcomes, they constructed an empirical analysis of different aggregations. Several well-known algorithms from the literature, such as support vector machines, decision trees, instance-based learning, and rule-based systems, are utilized in the experimental study. The results demonstrated the usefulness of binarization strategies in relation to the baseline classifiers, as well as the dependence of the aggregation strategy on the baseline classifier.

#### 2.5.4. Mixture of Expert Models

In [15], researchers explore numerous types of spectrograms in order to highlight various genre-specific properties for the MGC challenge. This research presents a mixture of expert (MoE) system to exploit all of these characteristics jointly. It is possible to derive a set of MGC models using different spectrogram-based statistics. Then, each model is treated as its expert. Consequently, a neural mixture model is introduced to collect and synthesize the expert models’ predictions and then generate a final prediction for a specific piece of music. They propose a neural-based MoE system, which can dynamically decide the weighting factor for each expert system to improve the performance of the MGC task.

In another similar domain, with minimal data available for the challenge of fine-grained recognition, it is impossible to develop diverse expertise by dividing the data. In [16], they combined an expert gradually enhanced learning technique with a Kullback–Leibler divergence-based constraint; they increased the variety among experts to address the challenge. The technique sequentially learns and adds new experts to the model based on the prior knowledge of previous experts. In contrast, the added constraint compels the experts to provide various prediction distributions. This compels the experts to understand the work from various angles, enabling them to develop expertise in a variety of subspace problems. Experiments indicate that the resulting model increases classification performance and reaches state-of-the-art performance on many fine-grained benchmark datasets.

## 3. Proposed Meta-Learning Techniques

The primary objective of this study is to test and evaluate several neural-network-based meta-learning strategies for solving the multiclass problem by combining ensemble learning, mixture of experts, one-vs.-rest decomposition, Bayes rule, and opinion aggregation approaches. By doing that, we try to address traditional multiclass methods’ drawbacks. At first, with the Bayes-based approach, we try to address the need to train either on different datasets on similar object types, but with diverse specifications or different acquisition protocols, or train on the same hierarchical dataset, but with different labels. Furthermore, our methods substantially enhance the database with unbalanced classes using another dataset of specific classes. In addition, we try to enhance the prediction power of one class that is weak in multiclass classification by using multiclass decomposition methods. In the following, we address techniques that are either part of the overall family of meta-learning techniques or a combination of several techniques. In particular, we focus on the image classification proble;m thus, “Base Learners” appearing in Figure 2 will represent “Classifiers” for the following proposed methods.

### 3.1. Ensemble Approach Based on Bayes Theorem

The first concept implemented is based on Bayes’ theorem. Specifically, this approach is an ensemble learning technique to enhance the prediction accuracy of an image classification task by utilizing the additional information gained by individual sub-class classifiers. The main idea is that useful information could be hidden in the relations formulated by the multiclass and the sub-class classifiers.

In detail, this technique can be applied when there is a classification problem with *n* different classes $C_{i}\in \left\{C_{1},\ldots ,C_{n}\right\}$ and *n* individual experts sub-class classifiers who can further divide each class $C_{i}$ into new sub-classes $Sc_{i}\in \left\{Sc_{1},\ldots ,Sc_{m}\right\}$, where *m* represents the number of sub-classes for the class $C_{i}$. Each sub-class classifier is implemented as a multiclass classifier which further divides class $C_{i}$ into *m* new *sub-classes*. The final classification probability $P_{Bayes}\left(C_{i}\right)$ is calculated for each initial class $C_{i}$ based on the Bayes theorem. As we will explain later in detail, $P_{Bayes}\left(C_{i}\right)$ is calculated by combining the posterior probability obtained as an independent observation from the expert sub-class classifiers and the output of the baseline multiclass classifier.

At first, we process the information gained by each expert sub-class classifier. In detail, each sub-class classifier $Sc_{i}$ (which is an expert for class $C_{i}$) returns a list of numbers in [0, 1] corresponding to the probabilities that an image belongs to any one of $Sc_{i}$’s sub-classes. Let $P_{Sc_{i}}\left(j|Ci\right)$ denote one such probability (namely the probability assigned by the classifier that the image belongs to its sub-class *j*). We pick the highest such $P_{Sc_{i}}\left(j|Ci\right)$ probability, and term its corresponding sub-class as $M_{i}$, as shown in Equation (Equation 2). Intuitively, $M_{i}$ is the sub-class of $C_{i}$ “favored” by the sub-class classifier $Sc_{i}$ as the most probable for the input image to belong to. Notice that $M_{i}$ is an observation corresponding to the likelihood $L(C_{i}|M_{i})$, which means that the higher the $M_{i}$ is, the higher the probability that an image belongs to class $C_{i}$.
(2)$$M_{i}=argmax\left(P_{Sc_{i}}\left(j|C_{i}\right)\right),\;\;\;j\in \left\{1,\cdots ,m\right\}$$
(3)$$\sum_{j=1}^{m}P_{Sc_{i}}\left(j\right)=1$$

We are now ready to use Equation (Equation 4) below to calculate what we term the “Bayes probability”. Equation (Equation 4) has a similar form to the Bayes rule (posterior ∝ likelihood · prior). There, the output $P_{multi}\left(C_{i}\right)$ of the baseline multiclass classifier denoting the probability assigned by that classifier to the image belonging in class $C_{i}$, plays the role of the prior, while $P(M_{i}|C_{i})$ returned for $M_{i}$ by the $Sc_{i}$ sub-class classifier (as explained above) plays the role of the likelihood, and $P_{Bayes}\left(C_{i}\right)$ is the posterior corresponding to the probability calculated by the middle level of our Bayes-theorem-based ensemble that the image belongs in class $C_{i}$.
(4)$$P_{Bayes}\left(C_{i}\right)=P\left(M_{i}|C_{i}\right)\cdot P_{multi}\left(C_{i}\right)$$

After calculating the class probabilities $P_{Bayes}\left(C_{i}\right)$ for each class $C_{i}$, these are used as inputs to a generalizer, which outputs a final prediction $P(C_{i})$ denoting the probability that an image belongs to each class $C_{i}$. Essentially, this generalizer is acting as a decision maker, which is trained to classify images into the *n* classes from the initial classification task, given only the $P_{Bayes}$ probabilities as shown in Figure 3.

The novelty of this approach lies in two parts. First, on using multiple unrelated multicenter datasets to train the expert sub-class classifiers, and second on how it calculates the final generalizer level inputs using the baseline multiclass classifier and the Bayes theorem. For the training of the generalizer, every image of the initial training dataset is fed to every classifier (sub-class classifiers and multiclass) to produce the $P_{Bayes}\left(C_{i}\right)$ values for every class $C_{i}$. Finally, we calculate the training loss by comparing the finally predicted $argmax(P\left(C_{i}\right))$, where i∈{1,⋯,n}, with the ground-truth class C.

However, this approach has one major drawback that limits it from being used by many image classification applications. Specifically, only a part of image classification problems have sub-class datasets available, which means that the individual datasets should be created from scratch or explored in detail to derive sub-classes beforehand; otherwise, this approach will not be applicable. Furthermore, another case where our Bayes-rule-based approach can be applied efficiently is the hierarchical (multi-label) datasets, where the additional labels can be used to define the sub-classes.

### 3.2. Mixture of Experts: Case of One-vs.-Rest Classifiers

The mixture of experts (MoE) is a popular and challenging combination strategy for improving machine learning performance. Mixture of experts is based on the divide-and-conquer approach, which divides the problem space among a few neural network experts and is regulated by a gating network. There exist different ways of partitioning the problem space between the experts. In brief, it entails breaking down predictive modeling tasks into sub-tasks, training an expert model on each, constructing a gating model that learns which expert to rely on based on the predicted input, and aggregating the predictions.

Although the technique was first published with neural network experts and gating models in mind, it can be applied to any form of a model. As a result, it addresses stacked generalization and belongs to the meta-learning class of ensemble learning approaches [30].

In detail, this technique can be applied in any multiclass classification problem with *n* different classes $C_{i}\in \left\{C_{1},\ldots ,C_{n}\right\}$. Its difference from the previous approach lies in calculating the final probability for the classes. Instead of relying solely on a neural network design in a conventional manner, we also employ a one-vs.-rest strategy. It requires partitioning the multiclass dataset into *n* binary classification problems, one for each distinct class. Then, a binary classifier is trained on each binary classification problem, and predictions are generated by combining the single binary class classifier with the conventional multiclass method.

The novelty of this proposed approach lies in how we calculate the final individual weights for each expert based on the output of the multiclass classifier. The final classification probability for each initial class $C_{i}$ is calculated using a combination based on the multiplication of: (a) the probability which is produced by the *single-binary* class classifiers for each individual binary problem and (b) the outputs of a baseline model (*multi-classifier*) acting as a gating function that is trained to solve the initial problem. In this mixture of experts approach, we use the initial data to create *n* new training datasets containing all the images of the initial dataset, depending on the class $C_{i}$ the images belong to. Specifically, each new dataset has two classes, the first resembling the case when an image belongs to class ${}_{i}$ and the second when an image does not belong in that class. In contrast with the previous Bayes-theorem-based approach, where the training datasets could have been different than the initial training dataset.

Specifically, we obtain from the baseline multiclass classifier the class probability $P_{multi}\left(C_{i}\right)$ and the probability $P_{yes_{i}}\left(C_{i}\right)$ from each single-binary classifier, as shown in Figure 4. Each expert is a single binary expert classifier $P_{yes_{i}}\left(C_{i}\right)+P_{no_{i}}\left(C_{i}\right)=1$, where $P_{yes_{i}}$ is the probability that an image belongs to the class $C_{i}$ and $P_{no_{i}}$ is the probability that an image does not belong to the class $C_{i}$ according to the single binary expert classifier *i*.
(5)$$P\left(C_{i}\right)=P_{multi}\left(C_{i}\right)\cdot P_{yes_{i}}\left(C_{i}\right)$$
(6)$$C_{index}=argmax_{index}\left(P\left(C_{i}\right)\right),\;\;\;i\in \left\{1,\cdots ,n\right\}$$

After calculating the class probabilities $P_{x}\in P\left(X=C_{1}\right),\ldots ,P\left(X=C_{n}\right)$ for each class using the formula in Equation (Equation 5), we then find the final classification result by searching for the probability $P(C_{i})$ with the highest value, as shown in Equation (Equation 6). Then, using the index, we find the predicted class $C_{index}$.

### 3.3. Proposed Combination Strategies

In this section, we put forward two stack generalization variants that follow the one-vs.-rest architectural paradigm. Stacked generalization is a method for lowering the generalization error rate of distinct generalizers. Stacked generalization operates by inferring the biases of the generalizer(s) relative to a given learning set. This deduction continues by generalizing in a second level whose inputs are (for instance) the opinions of the base learners when taught with a portion of the learning set and attempting to predict the remainder and whose output is (for instance) the correct guess. When employed with numerous generalizers, stacked generalization can be viewed as a more complex variant of cross-validation, as it employs a more sophisticated mechanism for merging the various generalizers than cross-validation’s simple winner-takes-all approach.

In stacking, an algorithm takes the outputs of sub-models as input and tends to learn the optimal way to combine the input predictions to obtain a more accurate output prediction. It may be useful to view the stacking process as having two levels, as depicted in Figure 2: level 0 and level 1.
Level 0: The training data are the inputs to the level 0 learners. Then, the level 0 base learners learn to make predictions for either the initial task or a specified sub-problem, depending on the training data.Level 1: The predictions produced by the level 0 learners are used as input to train the single level 1 meta-learner, with the intention to learn from this data to generate the final predictions.

Using a neural network as a meta-learner is usually advantageous when there are multiple base learners. Specifically, sub-networks can be embedded within a larger multi-headed neural network, which then learns how to combine the predictions from each input base learner optimally. It allows the stacking ensemble to be considered as a single large model. This approach has the advantage that the outputs of the base learners are provided directly to the meta-learner. Moreover, if desired, it is also possible to update the weights of the base learners in conjunction with the meta-learner model.

As commonly used in the literature, “one-vs.-rest” refers to a generic classification paradigm that employs binary methods for multiclass classification. It involves approaching the multiclass problem via the prism of multiple “binary” classifications. Then, a dedicated “binary” classifier (which may use a sigmoid function for classification into the most probable of the two classes) is trained on the original dataset and solves a “binary” classification problem. The final predictions are typically made using the most confident classifier or a simple, usually absolute, majority rule. The machine-learning algorithms established for binary classification can be extended to handle multiclass classification issues, which is one of the advantages of this technology. The downside is that the classifiers become unbalanced when the training data contain a disproportionate number of negative examples compared to positive examples. In our approaches, we exploit the benefits of such binary classifiers at the base level while enhancing their prediction power using “knowledge” at the layer of meta-learner.

Specifically, we combine properties of the stacked generalization technique schemes to provide novel improvements in the multiclass problem while addressing the diversity of classes and datasets. More specifically, in stack ensemble learning, the meta-learner can and should convey knowledge about the prediction power of base classifiers or the peculiarities of the different training datasets underlying the training of each base classifier. This is precisely the aim of our proposed schemes, which are introduced and discussed in the following.

#### 3.3.1. Stack Generalization (IP-Networks Only)

The first one-vs.-rest method we propose uses only the trained experts to solve the initial multiclass problem. Specifically, the original problem has been decomposed into *n* separate binary problems following the one-vs.-rest scheme, implemented with *n* separate binary classifiers, also mentioned in this work as “experts” or “Independent Parallel” classifiers (IP), as shown in Figure 5.

In this stack generalization variation, we create *n* new binary training datasets to train the single binary classifiers. The training data are fed to each expert with the entire image dataset and output two probabilities—one probability for the positive and one for the negative outcome. Then, the outputs of the individual experts are used as inputs to train a generalizer that learns to solve the initial multiclass problem. The final prediction produced by the generalizer is in the form of a list of probabilities for each class, summing to one (softmax layer).

This method’s performance depends on the individual experts’ prediction accuracy. However, as it will be shown later, stacked generalization methods manage to improve the base multiclass performance in every case regardless of the accuracy of the experts.

#### 3.3.2. Stack Generalization (IP-Networks + Multiclass CNN )

Like the previous method, the initial problem is divided into *n* smaller problems using binary classifiers, where *n* is the number of the original classes. The only technical difference with the previous method is that the generalizer uses extra information from the output of a baseline multiclass classifier, as shown in Figure 6.

This diagram is unrelated to mixture of experts in Figure 4 in two ways: first, in the manner in which the output of the multiclass classifier is employed, and second, in the presence or absence of a meta-learner (generalizer). In Figure 4, the output of the multiclass classifier serves as a gating function to regulate the participation of the single binary expert classifiers, whereas in this case, it is used as additional information as input to the meta-learner. Regarding the use of the meta-learner in Figure 6, it is an integral component of this method to produce the final class by combining the outputs of the independent classifiers and those of the multiclass.

In general, the addition of extra features in a classification task is usually advantageous and can significantly help to capture other aspects of the problem that previously were not completely observed. For instance, the IP-networks-only method cannot fully comprehend the broader picture and, more precisely, the relations between the classes other than the one the expert is trained in. By contrast, a baseline multiclass model is mainly trained to understand these inter-class relationships. Hence, adding the baseline multiclass model, along with the IP experts and the meta-learner (generalizer), is expected to improve the classification accuracy of the initial task significantly.

## 4. Experimental Evaluation

This section demonstrates the experimental process we used to evaluate the proposed methods’ performance. Initially, we discuss the experimental setup we used in three parts, the first being about the datasets we experimented with, the second the configuration, and the last the evaluation metrics. Right after, we present a detailed analysis by comparing the results of each proposed method along a baseline approach.

### 4.1. Experimental Setup

Initially, the experimental framework’s design and the configurations used are demonstrated. We start by presenting the specific datasets used to test the newly suggested methods, then we discuss the design of every test case, and finally we explain the reasons behind selecting the evaluation metrics. The main goal of this experimental framework is to provide precise and comprehensive results regarding the classification performance of the methods mentioned above.

#### 4.1.1. Datasets

Three datasets were used for the performance evaluation of this study. In every case, a stratified randomly divided 70% training set and 30% validation set split were applied as shown in Table 1. The datasets were chosen to demonstrate the adaptability of the various techniques to a wide range of situations using a variety of use cases. First, we evaluate two distinct subsets of the *herbarium* dataset with respect to the number of samples in each class and the balance of the datasets. This scalable increase in the number of classes in a hierarchically balanced dataset demonstrates the general case in the multiclass classification domain. In the case of UTKFACE, we deal with a highly imbalanced dataset with real-time applications in cutting-edge applications such as facial recognition. Finally, we test the case of building a new enhanced dataset by combining photos from two unrelated datasets to form a standard image database to evaluate the proposed methods’ performance in such conditions.

The first dataset we used in this work was the herbarium 2022 [31]. Herbarium 2022 is a hierarchical dataset that classifies around 80,000 images of herbs into a three-label (taxonomy) system. The first distinction level is the family, the next level is the genus, and the last is the species. In detail, herbarium contains around 250 families, 2500 genera, and 15,501 unique herbarium specimens. The initial problem linked with this dataset is to test the classification performance on a test dataset (with different images from similar plants as the training set) based on the unique ID of each plant species. As it can be understood, this problem’s size and complexity can make it especially hard to solve and even moreso when there is already the problem of class imbalance (each unique species can have only 5–80 training images). However, there are not many hierarchical datasets available, so for this research, two batches of herbarium families that make up the first two experimental datasets were distinguished. Specifically, the first dataset was formed using four families, and the second by using eight families and their related genera (sub-classes) to compose the first two experimental datasets.

The second dataset used in this work was the UTKFACE [32] Large scale face dataset. This dataset is a large-scale, multi-decade face dataset with an age range from 0 to 116 years old. In addition, the dataset contains approximately 20,000 face images with age, gender, and ethnicity annotations. The images exhibit a wide range of poses, facial expressions, illumination, occlusion, and resolution. This dataset may be utilized for various applications, including face identification, age estimates, age progression/regression, and landmark localization. The scope of this particular dataset is for image classification. For this study, the race categories, divided into the following five categories: (a) White, (b) Black, (c) Asian, (d) Indian, (e) Others (such as Hispanic, Latino, Middle Eastern) were treated as the level 0 classes, and for each class, the corresponding “binary” gender categorization was used as level 1.

The third dataset proposed in this experimental setup tackles the important “recyclable waste classification” problem. This dataset was created by images extracted from a selected subset of TrashBox [33] in combination with images from a subset of the Trashnet [34] dataset. TrashBox is a hierarchical dataset that has two labels, with the first one showing the material (e.g., plastic, paper, metal) and the second showing the type of item (e.g., newspaper, beverage can, …). In contrast, Trashnet is a simple multiclass dataset having six distinct classes with images of items from different materials (e.g., plastic, paper, metal, cardboard, …).

In this new particular dataset, there are three classes at level 0 (plastic, paper, metal) with the following sub-classes for each at level 1:Plastic: divided into plastic bags, plastic bottles, plastic containers, plastic cups, and cigarette butts.Paper: with sub-classes newspaper, paper cups, simple paper, tetrapak, and cardboard.Metal: with sub-classes beverage cans, construction scrap, metal containers, other metal, and spray cans.

The final blended dataset has around 10,000 unique images with two levels of annotations.

#### 4.1.2. Configuration and Specifications

As known in the machine learning community, it is not an easy task to determine when a specific classification approach performs well just by looking at the metrics of the specific implementation alone. The simplest solution would be to create a baseline approach and compare it to the new implementations, while ensuring a set of rules is applied. In this study, the ResNet50 [35] model was widely used to train all the classifiers of all mentioned methods, while using the same training parameters (learning rate, batch size) to ensure a fair comparison between the methods. The learning rate parameter is one of the factors that determines the convergence speed that directly affects the training performance. The learning rate was set to 0.01 with Adam’s policy [36]. Additionally, a baseline ResNet50 multiclass classifier was trained for every dataset, using the same training parameters as the rest of the proposed methods to have a solid starting point to measure the performance improvement of every approach. Thus, ResNet50 networks were used as base classifiers, while the generalizers had a different, more straightforward form. Furthermore, all *generalizers* had the same architecture (Figure 7) and were designed to be as simple and lightweight as possible. Specifically, it comprised two fully connected layers with 256 nodes and a relu activation each, while having a batch normalization layer in between, followed by a dropout layer, and finally one more fully connected layer with softmax as an activation function, with a number of nodes equal to the number of classes to classify.

#### 4.1.3. Evaluation Metrics

The selection of the appropriate evaluation metrics can be very challenging sometimes. Therefore in this study, four of the most used metrics were selected. The macro average [37] of the top four performance metrics will be compared to determine how effective the proposed methods are. The most common evaluation metric is the overall accuracy of a multiclass classifier. The average multiclass classifier accuracy (Equation (Equation 7)) is calculated as the division of the number of accurate predictions and the total number of predictions.
(7)$$Accuracy=\frac{Number\;of\;correct\;predictions}{Total\;number\;of\;predictions}$$

However, the information provided by the average accuracy can be misleading, especially in problems with more classes; hence other metrics should be considered too to determine whether there are significant improvements in the overall performance of a classifier. For this reason, we also measured the effectiveness of our proposed methods in terms of precision, recall, and F1 score.

In a multiclass classification problem with *n* different classes $C_{i}\in \left\{C_{1},\ldots ,C_{n}\right\}$, we define as $tp_{i}$ the number of true positive predictions for $C_{i}$, i.e., the number of times the multiclass classifier correctly predicted the positive class $C_{i}$. Additionally, $fp_{i}$ is the number of false positives for $C_{i}$, defined as the outcome where the model incorrectly predicts class $C_{i}$. By contrast, $fn_{i}$ are the false negative predictions for $C_{i}$.

Equation (Equation 8) shows the precision of a classifier which represents the proportion of positive predictions that were actually correct. Since the problem is multiclass, the averages of the precision $\frac{tp_{i}}{tp_{i}+fp_{i}}$ of every class $C_{i}$ are considered.
(8)$$Precision=\frac{\sum_{i=1}^{n}\frac{tp_{i}}{tp_{i}+fp_{i}}}{n}$$

Respectively, the recall metric, displayed in Equation (Equation 9), shows the proportion of actual positives that were identified correctly. Here again, to calculate the average recall, the individual recall of every class $C_{i}$ is assessed.
(9)$$Recall=\frac{\sum_{i=1}^{n}\frac{tp_{i}}{tp_{i}+fn_{i}}}{n}$$

Another helpful metric is the F1 score (Equation (Equation 10)) because it combines a multiclass classifier’s precision and recall.
(10)$$F1_{score}=\frac{2\cdot Precision\cdot Recall}{Precision+Recall}$$

By using these metrics, we can better determine when one of the developed methods improves the overall performance or just a specific portion of the problem, while not overlooking the main classification task.

### 4.2. Results and Analysis

In this section, a detailed analysis of the experimental results of the proposed methods is presented. These methods were applied and tested using four different paradigms with varying classification difficulty.

Initially, the specifications of every architecture will be discussed according to Table 2. The first column displays the total trainable parameter count of the proposed approaches in millions, while the second column demonstrates the floating point operations (FLOPs) required to complete an inference step in billions. The *n* parameter denotes the number of classes of the original multiclass problem. The parameter count and the FLOPs are proportional to the number of classes *n* because in our decomposition-based approaches, every new class requires an individual binary or sub-class classifier. The third column specifies the time in milliseconds (ms) needed for a single inference step to be conducted. To obtain robust insights about the time per inference step, we run 30 repetitions of the same 200 images for all architectures in a PC equipped with an Intel i7-9700 CPU, 32 GB of RAM, and an NVIDIA RTX 2080 SUPER 8GB GPU. The last column presents the depth of every neural network, counting only the layers with trainable parameters.

As expected, the baseline multiclass architecture, in our case ResNet50, was the most lightweight in terms of parameter count and time per inference step. In particular, the time per inference step presented was generated from running the different methods in the five-class UTKFACE dataset. Interestingly, the baseline multiclass classifier was about six times faster than all the other methods except the IP-only method, which was only close to five times faster. By contrast, the depth of the architectures remained similar since the methods we propose make the overall architecture grow wider and not deeper, which can also be a practical feature because many operations can be performed in parallel, hence reducing the inference time step when run on a high-end machine. Furthermore, the number of parameters and the floating point operations of the proposed methods increase linearly with respect to the number of classes *n* of the initial multiclass classification problem. Thus, a multiclass classification problem with 1000 classes, such as ImageNet [38], would require considerable storage space; however, the accuracy improvement that can be accomplished in ImageNet using our proposed methods has not yet been tested.

Even though the inference time increased, multiclass problems using the proposed architectures can still be solved in real-time. Therefore, this slight increase in inference time is only an insignificant trade-off—for multiclass problems with a small number of classes—compared to the classification accuracy improvement that can be achieved, which will be demonstrated in the following tables.

First, the results from the four-class herbarium dataset are shown. As depicted in Table 3, the classification task is not that complicated; hence, the overall baseline performance is quite high, having 95.32% accuracy, so the potential performance improvement gap cannot be as significant. However, every proposed method managed to increase every metric that is monitored when compared to the baseline. In detail, the Bayes-rule-based method achieved +0.14% accuracy improvement, the mixture of experts showed a +0.82% improvement, the modified stacked generalization (IP-only) by +0.94%, and the modified stacked generalization (IP with multiclass) managed to increase the classification accuracy by +1.09%.

The proposed implementations were tested on datasets of varying improvement gaps to produce solid results. The four-class herbarium dataset resembles a more straightforward scenario where the classes and the subcategories of each class are well balanced. In contrast, the eight-class herbarium resembles a more complex scenario, where the first dataset contains images from four additional classes, eight in total, and the subcategories are slightly unbalanced. Furthermore, the first dataset has class peculiarities (especially in the eight-class formulation) that render the multiclass distinction and classification quite challenging. Here, our proposed methodologies show their advantages in prediction power.

Specifically, as shown in Table 4, the baseline accuracy was 80.59% with the Bayes method having a +1.09% improvement, the mixture of experts method having +3.12%, the stacked ensemble (IP-only) method by +3.37%, and the IP with a multiclass method having the most remarkable improvement of +5.89%. In this case, too, all the proposed methods achieved better overall results than the baseline, showing that the proposed methods can be even more helpful in more complex classification tasks.

The last two experiments were based on well-designed datasets, which exemplify the properties of different classes. For this reason, all classifiers’ performance was high and quite comparable. UTKFACE was the third dataset tested, and it resembles an even more challenging classification scenario, where the optimal solution is not that clear. In detail (Table 5), the baseline implementation achieved only 72.77% accuracy, which means that the improvement gap could be quite large. In this case, the Bayes method managed to increase the classification accuracy by +2.09%, the IP-only method increased it by +3.28%, and the IP with the multiclass method by +4.64%. Finally, the mixture of experts achieved the most significant improvement of +5.87%. This experimental scenario proves that the most “stacked” model is not always the best. At the same time, a more elegant approach, such as mixture of experts without any meta-learners, can be better in specific use cases.

The last dataset tested represents an easier classification task, with the baseline achieving 94.14% accuracy, as depicted in Table 6. Additionally, The Bayes-theorem-inspired method achieved only a +0.2% accuracy improvement, the MoE method achieved a +0.94% improvement, and the stacked generalization (IP-only) a +0.16% increase. In contrast, the stacked generalization (IP with multiclass) improved significantly by +1.02%.

Overall in almost all four dataset cases, the Bayes method achieved the slightest performance improvement, varying from +0.12% in the stiffest case up to +2.09% in the UTKFACE test case, where the improvement gap was the greatest. The mixture of experts approach was surprisingly resourceful in the classification task of the UTKFACE dataset, achieving the most remarkable increase across all monitored metrics. However, this method performed exceptionally well in the other test cases too. The IP-only method outperformed the Bayes-theorem-inspired method in three out of four test cases and the MoE only in two out of four test cases. While this method never outperformed the most complex stacked generalization method, it proved to be a stable middle-ground between the two most well-performing approaches. Finally, the stacked generalization (IP with multiclass) method had the best metrics in every case except in the UTKFACE dataset, where it was the second-best approach. Overall, this method had the most stable performance, proving that in most cases, the more complex the stacked model, the more remarkable improvement it can achieve.

## 5. Conclusions and Future Work

In conclusion, we proposed four new classification approaches in this study and evaluated them on four distinct datasets representing use cases of varying difficulty. Specifically, we presented a series of novel approaches for combining the output of a decomposed multiclass (image) classification task under the umbrella of the one-vs.-rest framework for the opinion aggregation module. The first was a novel opinion aggregation mechanism that combines information derived from sub-class classifiers based on the Bayes theorem. The second was a novel design for the mixture of expert approaches that incorporates the knowledge of a multiclass classifier as a gating model. Lastly, we put forward two stack generalization variants with novel characteristics that follow the one-vs.-rest architectural paradigm.

In the end, all methods improved all monitored evaluation metrics when compared to the baseline multiclass classifier. Our goal in this study was not only to design an improved classifier but also to explore methodologies that can diversify small class differences to be able to derive specific sub-classes hidden in the data.

The performance of the newly proposed mixture of experts method, which does not employ any additional meta-learners (generalizers), was highly remarkable. In addition, the proposed Bayes-theorem-based approach demonstrated the quite remarkable fact that information gained by a second label, such as in the herbarium and UTKFACE cases, or by an entirely new dataset, such as in the waste classification problem, can provide researchers with new classification “*features*” to utilize. Specifically, in a future study, the impact of adding these new features to any classification task could be evaluated and quantified, thereby providing researchers with more information about the limitations of this approach. Furthermore, even though the performance of our stacked generalization methods with the generalizers was anticipated to be rewarding, this shows that traditional meta-learning is still a powerful tool to be considered in multiclass classification tasks.

## Figures and Tables

**Figure 1 sensors-23-00009-f001:**
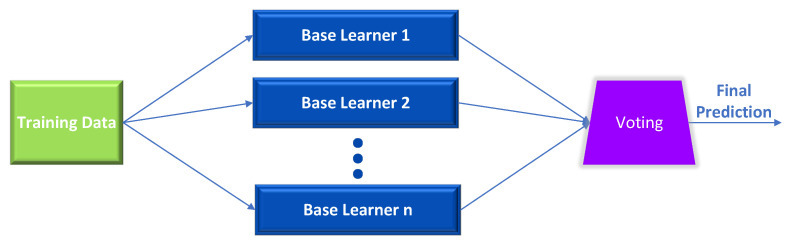
Ensemble learning voting architecture.

**Figure 2 sensors-23-00009-f002:**
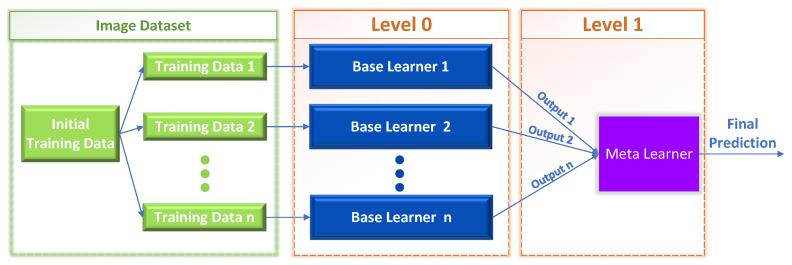
Stacked generalization architecture.

**Figure 3 sensors-23-00009-f003:**
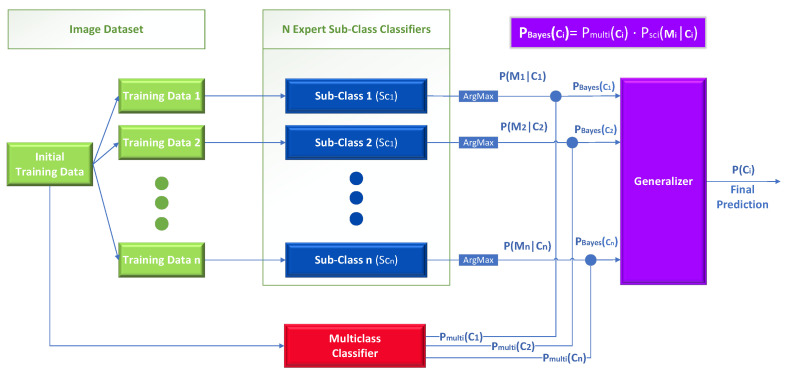
The architecture of the Bayes-theorem-based Ensemble approach.

**Figure 4 sensors-23-00009-f004:**
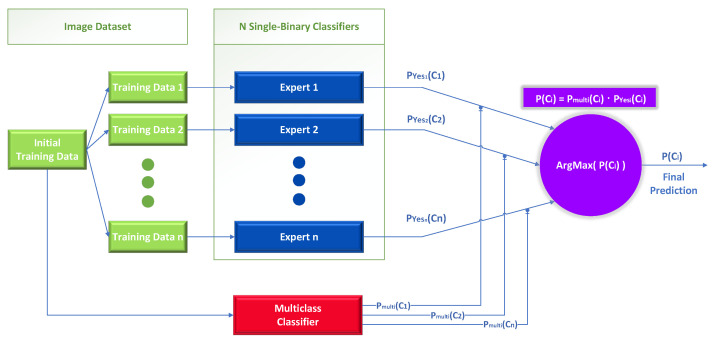
Mixture of experts; one-vs.-rest classification.

**Figure 5 sensors-23-00009-f005:**
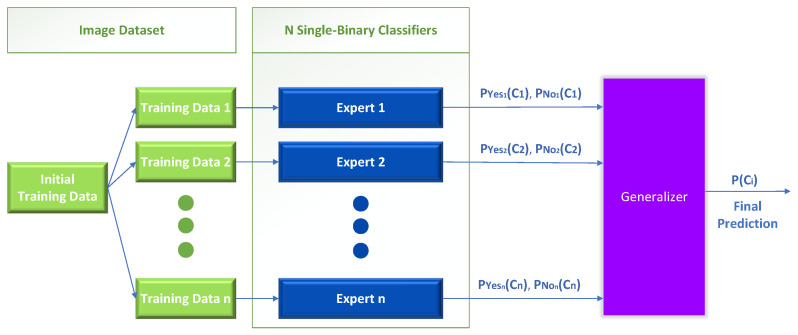
Meta-Learning, binary classifiers only.

**Figure 6 sensors-23-00009-f006:**
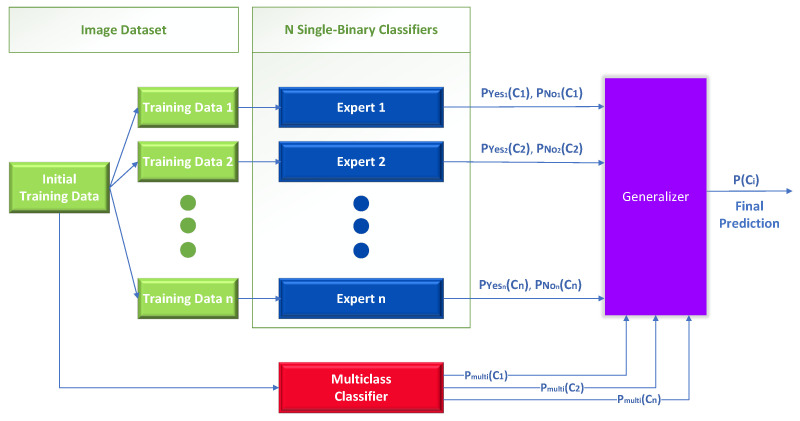
Meta-Learning, binary, and multiclass classifiers generalizer.

**Figure 7 sensors-23-00009-f007:**
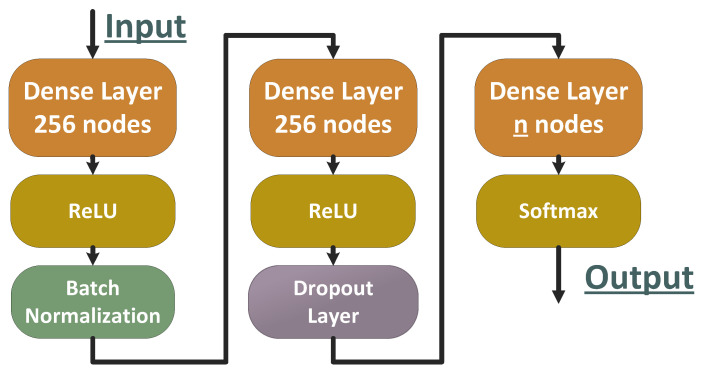
Network architecture of the *generalizers*.

**Table 1 sensors-23-00009-t001:** Summary description of the datasets used in the experiments.

Dataset	Number of Classes	Total Images	Distribution of Images in Classes
**four-class Herbarium**	4	19,728	4354|5720|3194|6460
**eight-class Herbarium**	8	38,535	4354|4346|5109|4357|4995|5720|3194|6460
**UTKFACE**	5	24,104	10,222|4558|3586|4027|1711
**Waste Classification**	3	8158	1909|4313|1936

**Table 2 sensors-23-00009-t002:** Specification comparison of every proposed architecture in terms of parameter count, FLOPs, inference time, and depth.

	Parameters (Million)	FLOPs (Billion)	Time (ms) per Inference Step (GPU)	Depth
**Baseline Multiclass**	24.7	7.75	27	107
**Bayes Method**	(n + 1) · 24.7	(n + 1) · 7.75	163	110
**Mixture of Experts (without meta learner)**	(n + 1) · 24.7	(n + 1) · 7.75	162	107
**Stacked Generalization (IP only)**	n · 24.7	n · 7.75	136	110
**Stacked Generalization (IP + multiclass)**	(n + 1) · 24.7	(n + 1) · 7.75	165	110

**Table 3 sensors-23-00009-t003:** Evaluation metrics for the four-class herbarium dataset. The highest accuracy metrics are highlighted using bold.

Four-Class Herbarium Dataset	Accuracy	Precision	Recall	F1
**Baseline Multiclass**	95.32%	94.79%	94.73%	94.76%
**Bayes Method**	95.44%	94.91%	94.88%	94.89%
**Mixture of Experts (without meta learner)**	96.14%	95.67%	95.70%	95.68%
**Stacked Generalization (IP only)**	96.26%	95.89%	95.52%	**95.69%**
**Stacked Generalization (IP + multiclass)**	**96.41%**	**96.14%**	**95.85%**	95.59%

**Table 4 sensors-23-00009-t004:** Evaluation metrics for the eight-class herbarium dataset. The highest accuracy metrics are highlighted using bold.

Eight-Class Herbarium Dataset	Accuracy	Precision	Recall	F1
**Baseline Multiclass**	80.59%	80.58%	80.45%	80.07%
**Bayes Method**	81.68%	81.49%	80.89%	81.13%
**Mixture of Experts (without meta learner)**	83.71%	83.58%	83.56%	83.32%
**Stacked Generalization (IP only)**	83.96%	83.48%	83.39%	83.38%
**Stacked Generalization (IP + multiclass)**	**86.48%**	**86.08%**	**86.00%**	**86.01%**

**Table 5 sensors-23-00009-t005:** Evaluation metrics for the UTKFACE dataset. The highest accuracy metrics are highlighted using bold.

UTKFACE Dataset	Accuracy	Precision	Recall	F1
**Baseline Multiclass**	72.77%	65.55%	67.33%	66.04%
**Bayes Method**	74.86%	67.64%	66.68%	67.00%
**Mixture of Experts (without meta learner)**	**78.64%**	**71.09%**	**71.27%**	**71.07%**
**Stacked Generalization (IP only)**	76.05%	70.20%	68.76%	69.06%
**Stacked Generalization (IP + multiclass)**	77.41%	70.69%	70.06%	70.19%

**Table 6 sensors-23-00009-t006:** Evaluation metrics for the waste classification dataset. The highest accuracy metrics are highlighted using bold.

Waste Classification Dataset	Accuracy	Precision	Recall	F1
**Baseline Multiclass**	94.24%	93.04%	93.94%	93.46%
**Bayes Method**	94.44%	93.34%	93.94%	93.63%
**Mixture of Experts (without meta learner)**	95.18%	**94.14%**	94.81%	**95.18%**
**Stacked Generalization (IP only)**	94.40%	93.07%	94.20%	93.59%
**Stacked Generalization (IP + multiclass)**	**95.26%**	94.13%	**94.99%**	94.55%

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
