# Peer review of "Novel Meta-Learning Techniques for the Multiclass Image Classification Problem"

_sensors, 2022, doi:10.3390/s23010009_

Round 1

Reviewer 1 Report

In this manuscript, authors introduce four different approaches for optimizing the ensemble phase of multiclass classification: Mixture of Experts scheme, technique for combining learner-based outcomes that relies on Bayes’ theorem and two additional methods based in the stack generalization. 

I consider the idea to be innovative and worth publishing, however, some aspects should be addressed:

1) I suggest merging Introduction and Background sections, rearrange it to more clearly describe the state-of-the-art (this part seems explained in the 5th section), and stress the scientific contributions of your work in relation to the shortcomings of the existing multiclass classification methods. The fifth section should be rearranged and incorporated into the Introduction.

2) A part of the Background section should be moved to Methodology (Proposed Meta-Learning techniques) in order to adequately support the description of your novel approaches.

3) In the Results section (Experimental evaluation) authors only use ResNet50 as benchmark classifier, I suggest adding few more benchmark algorithms.

I suggest taking a look at:

La Cava, W., Silva, S., Danai, K., Spector, L., Vanneschi, L., & Moore, J. H. (2018). Multidimensional genetic programming for multiclass classification. Swarm and Evolutionary Computation. doi:10.1016/j.swevo.2018.03.015

Author Response

1)I suggest merging Introduction and Background sections, rearrange it to more clearly describe the state-of-the-art (this part seems explained in the 5th section), and stress the scientific contributions of your work in relation to the shortcomings of the existing multiclass classification methods. The fifth section should be rearranged and incorporated into the Introduction.

  • We have revised our text. In detail, we have moved the former Section 5, “Related Work” to the end of the Background section, so now it will make more sense to be there. We also added a concise paragraph about the state-of-the-art related work in the introduction.

2) A part of the Background section should be moved to Methodology (Proposed Meta-Learning techniques) in order to adequately support the description of your novel approaches.

  • We have moved a part of the background regarding “Stacked generalization”, to the 3.3 “Proposed Combination Strategies” to introduce better the stacked generalization methods we proposed.

3) In the Results section (Experimental evaluation) authors only use ResNet50 as benchmark classifier, I suggest adding few more benchmark algorithms.

  • Unfortunately, time constraints for updating the submission make it impossible to perform systematically these additional experiments; however, we can report here that preliminary results when we experimented with different architectures, such as EfficientNetB0 and DenseNet201 for specific datasets, show no significant changes in behavior or performance.

Reviewer 2 Report

By reviewing and discussing four approaches for optimizing the ensemble phase of multiclass classification, the submitted article tries to assess the efficiency of the decomposition-based methods and to offer some minor improvements to the meta-learning level where the outputs of the many individual models are combined. Generally speaking, the level of novelty in the presented work is not high as it just reviews some of the well-known methodologies, and at some point, it tries to have more contributions by adding a very limited level of innovations. But the topic is very interesting and to the point. In other words, the submitted research discusses a REAL concern rather than just creating a new problem (that may never happen in reality!) and suggesting some solutions to solve that. Therefore, I believe, the work deserves to be read and regarded in the community. The language is fine as the paper is readable and easy/clear to follow but still needs revision. The structure, however, needs to be thoroughly revised.

-         Please add adequate discussions about the complexity of each approach in terms of the computational time and so on, and report the benchmarking results in tables. This may give a better idea to the readers how to choose the best method, especially regarding the fact that the differences between the four methods were not that significant in terms of accuracy, precision, recall, and F1.  

-         The structure of the paper needs serious revision. More specifically, section 5 (related work) should either merge with the introduction section or come as a separate section, but surely before introducing the proposed method and reporting the results.

-         There are several typos and language issues here and there, please proofread the manuscript to fix such issues.

-         If possible, I suggest slightly shortening the Abstract by a few sentences.

-         Equations 8 to 10 and/or their explanations need revision. First, what does subscript M mean in Precision-M and Recall-M? Are they necessary to be added to Precision and Recall. If yes, please explain them after the equations. Also, in the explanations below the equations, what is n? That should be replaced with N? the definition of Ci is missing as well. since the definition of fpi and tpi is based on Ci, I suggest first introducing Ci.

-         The two fully connected layers in Figure 7 are with 256 nodes, but the way this number is overlaid on the blocks is a bit confusing. I suggest writing the numbers in a vertical manner, like the way RELU and SOFTMAX are written in this figure.

-         Please use appropriate headings for the sections and sub-sections. More specifically, the one for the references (at the end of the article) is missing. 

Author Response

  1. Please add adequate discussions about the complexity of each approach in terms of the computational time and so on, and report the benchmarking results in tables. This may give a better idea to the readers how to choose the best method, especially regarding the fact that the differences between the four methods were not that significant in terms of accuracy, precision, recall, and F1.  
  •  We have added additional information about the “inference time step (ms)”, Gflops, depth, and the number of parameters of every architecture we propose (see Τable 2), and also discussed them at the start of the  “4.2. Results and analysis” Section.
  1. The structure of the paper needs serious revision. More specifically, section 5 (related work) should either merge with the introduction section or come as a separate section, but surely before introducing the proposed method and reporting the results.
  •  We have merged parts of the former section 5 with the introduction and also moved all parts of the old section 5 to a new subsection in the Background. So, the “Related Work” will be introduced before discussing our proposed methods in Section 3.
  1. There are several typos and language issues here and there, please proofread the manuscript to fix such issues.
  •  We have taken some time to proofread and spellcheck our paper, so we hope there will not be many mistakes left.
  1. If possible, I suggest slightly shortening the Abstract by a few sentences.
  •  We have shortened the abstract, so now it should be more concise.
  1. Equations 8 to 10 and/or their explanations need revision. First, what does subscript M mean in Precision-M and Recall-M? Are they necessary to be added to Precision and Recall. If yes, please explain them after the equations. Also, in the explanations below the equations, what is n? That should be replaced with N? the definition of Ci is missing as well. since the definition of fpi and tpi is based on Ci, I suggest first introducing Ci.
  • We have rewritten some parts around Equations 8, 9, and 10 to explain them more accurately. 
  1. The two fully connected layers in Figure 7 are with 256 nodes, but the way this number is overlaid on the blocks is a bit confusing. I suggest writing the numbers in a vertical manner, like the way RELU and SOFTMAX are written in this figure.
  • We have replaced Figure 7 with a new figure that illustrates the generalizer architecture in more detail. 
  1. Please use appropriate headings for the sections and sub-sections. More specifically, the one for the references (at the end of the article) is missing. 
  • We have fixed this; thanks for noticing.

Round 2

Reviewer 1 Report

Accept in present form.